# A Chaos Theory Approach to Understand Neural Network Optimization

## Abstract

Despite the complicated structure of modern deep neural network architectures, they are still optimized with algorithms based on Stochastic Gradient Descent (SGD). However, the reason behind the effectiveness of SGD is not well understood, making its study an active research area. In this paper, we formulate deep neural network optimization as a dynamical system and show that the rigorous theory developed to study chaotic systems can be useful to understand SGD and its variants. In particular, we first observe that the inverse of the instability timescale of SGD optimization, represented by the largest Lyapunov exponent, corresponds to the most negative eigenvalue of the Hessian of the loss. This observation enables the introduction of an efficient method to estimate the largest eigenvalue of the Hessian. Then, we empirically show that for a large range of learning rates, SGD *traverses the loss landscape across regions with largest eigenvalue of the Hessian similar to the inverse of the learning rate*. This explains why effective learning rates can be found to be within a large range of values and shows that SGD implicitly uses the largest eigenvalue of the Hessian while traversing the loss landscape. This sheds some light on the effectiveness of SGD over more sophisticated second-order methods. We also propose a quasi-Newton method that dynamically estimates an optimal learning rate for the optimization of deep learning models. We demonstrate that our observations and methods are robust across different architectures and loss functions on CIFAR-10 dataset.

## 1    Introduction

An interesting observation from current deep learning research is that classification and regression accuracy gains seem to be achieved from the intricacy of the underlying models rather than the optimization algorithm used for their training. Actually, the de facto choice for the optimization algorithm is still the classic Stochastic Gradient Descent (SGD) algorithm (Robbins & Monro, 1951) with minor modifications (Duchi et al., 2011; Sutskever et al., 2013; Kingma & Ba, 2014). Even though several sophisticated second-order and quasi-Newton methods (Martens, 2010; Martens & Grosse, 2015; Berahas et al., 2019) have been introduced, first-order methods remain popular and none of them seem to outperform SGD with a carefully tuned learning rate schedule (Hardt et al., 2016). This indicates that SGD (or in general first-order methods) probably has some intrinsic properties that make it effective to optimize over-parametrized deep neural networks. Despite various attempts to explain such phenomenon (Chaudhari & Soatto, 2018; Keskar et al., 2016; Kleinberg et al., 2018), little is understood about the effectiveness of SGD over sophisticated second-order optimization methods.

In this paper, we argue that chaos theory (Sprott & Sprott, 2003) is a useful approach to understand the neural network optimization based on SGD. The basic idea is to view neural network optimization as a dynamical system where the SGD update equation maps from the space of learnable parameters to itself and describes the evolution of the system over time. Once the evolution is defined, the rich theory developed to study chaotic dynamical systems can be leveraged to analyze and understand SGD and its variants. In essence, chaos theory enables us to study the evolution of the learnable parameters (*i.e.*, the optimization trajectory) in order to understand the training behavior over large time scales (*i.e.*, number of iterations).

In particular, we focus on understanding the influence of the learning rate on the SGD optimization trajectory. First, by observing that the Lyapunov exponent of SGD is the most negative eigenvalue of the Hessian of the loss, we introduce an efficient and accurate method to estimate the loss curvature. Then, we empirically show that for a range of learning rate schedules, SGD traverses the optimization landscape across regions with largest eigenvalue of the Hessian similar to the inverse of the learning rate. This demonstrates that at a specific time step, performing SGD update is similar to performing a quasi-Newton step, considering only the largest eigenvalue of the Hessian of the loss. This, for the first time, sheds some light on the effectiveness of SGD over more sophisticated second-order methods and corroborates the observation that SGD robustly converges for a variety of learning rate schedules (Sun, 2019).

Furthermore, as pointed out in (LeCun et al., 1993), the inverse of the estimated curvature can be used as the learning rate when applying SGD to a new dataset or architecture. Hence, we can set up a "feedback" system where the quasi-Newton optimal learning rate is calculated dynamically based on the current largest eigenvalue of the Hessian (curvature), and the learning rate is consequently adjusted during the training, allowing a "parameter free" stochastic gradient descent optimization. The experiments are conducted on CIFAR-10 dataset to demonstrate that our observations are robust across a variety of models, including a simple linear model regression and more modern deep neural network architectures, trained with both cross entropy and mean square error loss functions.

## 2 Chaos Theory for Neural Network Optimization

In recent years, several papers have used dynamical systems to study theoretical aspects of deep learning optimization (Liu & Theodorou, 2019). Essentially, this is achieved by defining the optimization of deep neural networks as the evolution of parameters over time. In particular, a dynamical system progresses according to a map function that describes how the system evolves in a specific time step. In the case of deep neural network optimization, this map function is defined from the space of parameters into itself. By describing the system evolution using such a map function, it is possible to leverage the mathematical machinery of dynamical systems. For instance, viewing SGD as a discrete approximation of a continuous stochastic differential equations, allowed Li et al. (2017) and An et al. (2018) to propose adaptive SGD algorithms. Furthermore, dynamical systems enabled LeCun et al. (1993) to relate learning rate with the inverse of the local Hessian in a quasi-Newton optimization framework. Our paper also uses dynamical systems to study deep learning optimization, but differently from all methods above, we rely on chaos theory.

Chaos theory (Sprott & Sprott, 2003) studies the evolution of dynamical systems over large time scales and can categorize systems into chaotic or non chaotic. Under some simplifying but still general assumptions, chaotic systems are bounded and have strong dependence on the initial conditions. This means that chaotic systems evolving from different starting points that are within a relatively small region around a particular reference point, will diverge exponentially during the evolution process, where the amount of time taken for this divergence to happen is defined as the chaotic timescale. This chaotic timescale imposes a limit on our ability to predict the future state distribution of a dynamical system. In fact, the distribution of the future state, which have evolved for more than a few times the chaotic timescale, cannot be distinguished from random distributions, even when the system is fully deterministic. We apply concepts from chaos theory to improve our current understanding of the optimization of deep neural networks.

More specifically, we describe how to use standard chaos theory techniques to efficiently calculate the leading (positive and negative) eigenvalues of the Hessian of the loss function. With these eigenvalues we measure, in turn, the loss function curvature, which can be used to study the behavior of first-order optimization methods, such as SGD (Robbins & Monro, 1951). In particular, with this technique we formulate an explanation for the empirical robustness of SGD to the choice of learning rate and its scheduling function, and we investigate a method (based on quasi-Newton second order method) for dynamically finding the optimal learning rate during the optimization of deep neural networks. Such automated and dynamic estimation of optimal learning rate can lift a significant burden from the manual definition of learning rate schedules in deep learning optimization.

## 2.1 LYAPUNOV EXPONENTS

In chaos theory, the Lyapunov exponents define the divergence rate of infinitesimally close trajectories, and the inverse of the largest Lyapunov exponent is the timescale that corresponds to the onset of chaos into the system. Two arbitrarily close initial conditions generate two solutions that diverge with time. Under the assumption that the map function of the system is differentiable, if one observes this divergence for a short time window, it grows exponentially. If the initial divergence $\boldsymbol{q}(0)$ is made smaller, the time window can be made larger ($t \rightarrow \infty$). The largest Lyapunov exponent $\lambda$ is a measure of the growth of the divergence $\boldsymbol{q}(t)$ in the direction $\hat{\boldsymbol{q}}(0) = \boldsymbol{q}(0)/\|\boldsymbol{q}(0)\|$ with the largest growth ($\max_{\hat{\boldsymbol{q}}(0)}$) along the trajectory, as in

$$\lambda = \max_{\hat{\boldsymbol{q}}(0)} \lim_{t \rightarrow \infty} \lim_{\|\boldsymbol{q}(0)\| \rightarrow 0} \frac{1}{t} \log \frac{\|\boldsymbol{q}(t)\|}{\|\boldsymbol{q}(0)\|} \ . \tag{1}$$

In this paper, we rely on the local finite size Lyapunov exponent. In this context, local in time means that there is no limit to infinity for $t$ in equation 1 – instead, it is an average over a constant time window $t$. Finite size means keeping the difference in parameter space fixed as a small constant with $\|\boldsymbol{q}\| = \Delta q$ (*i.e.*, no limit $\|\boldsymbol{q}\| \rightarrow 0$ in equation 1). Using a finite size allows the study of the dynamic system at a specific spatial scale (for a comprehensive review, see (Cencini & Vulpiani, 2013)), corresponding to the eigenvalues of the Hessian of a spatially smoothed version of the loss (or equivalently, to the numerical second derivative with a finite delta). When this analysis is used to study the Hessian, this is equivalent to calculating the local numerical second derivative. We found empirically that the results do not depend on the $\Delta q$ parameter within a large range of values.

We will show in Sec. 3 that this timescale (*i.e.*, the Lyapunov exponent) corresponds to the most negative eigenvalue of the Hessian of the loss, when optimizing deep neural networks with SGD. Intuitively, a small difference in the initial condition will amplify exponentially in the directions with negative second derivatives and will dampen in directions with positive second derivatives. Empirically, we find that the chaotic timescales in effective training of deep neural networks are short (in the order of tens of iterations) when compared with the time of one epoch (*i.e.*, the total number of iterations in one epoch). We also find that there are multiple unstable directions throughout the training, *i.e.*, the system is hyper-chaotic.

## 2.2 LYAPUNOV EXPONENTS FOR GD AND SGD

In this section we derive the formula to compute the largest Lyapunov exponents for Gradient Descent (GD) following (Sprott & Sprott, 2003). We first show that the largest Lyapunov exponent corresponds to the most negative eigenvalue of the Hessian of the loss and provide an algorithm to efficiently compute it. This will be later extended to calculate the largest (or in general the top-$k$) eigenvalue of the Hessian in section 3. For simplicity of the exposition, in this section we initially consider the non-stochastic setting. Also, for the results of this section to hold, we assume that the Hessian of the loss does not change quickly through time, and it does not change quickly along the optimization trajectory compared to the chaotic time scale. These assumptions can easily be checked a posteriori, and we will show how to overcome this (potential) limitation in section 3.

Let $\boldsymbol{\theta}$ be the vector of learnable parameters of the deep neural network, $L(\cdot)$ be the loss function, and $\alpha > 0$ be the learning rate. The gradient descent step at iteration $t$ is written as:

$$\boldsymbol{\theta}_{t+1} = \boldsymbol{\theta}_t - \alpha \frac{dL(\boldsymbol{\theta}_t)}{d\boldsymbol{\theta}} \ , \tag{2}$$

where the update step $\Delta\boldsymbol{\theta} = -\alpha \, dL/d\boldsymbol{\theta}$. In the limit of small steps the formulation is equivalent to a Partial Differential Equation (PDE)

$$\frac{d\boldsymbol{\theta}}{dt} = -\alpha \frac{\partial L(\boldsymbol{\theta})}{\partial \boldsymbol{\theta}} \ . \tag{3}$$

Integrating equation 3 gives the evolution of the system, which is equivalent to training the neural network.

To compute the chaotic time scale (i.e. the inverse of the Lyapunov exponent), one needs to analyze the difference in evolution of GD at two arbitrarily close initial points. To this end, we consider a

small perturbation $\boldsymbol{q}_0$ added to the initial weights $\boldsymbol{\theta}_0$. For this perturbed starting point $\boldsymbol{\theta}_0 + \boldsymbol{q}_0$, the PDE becomes:

$$\frac{d(\boldsymbol{\theta} + \boldsymbol{q})}{dt} = -\alpha \frac{\partial L(\boldsymbol{\theta} + \boldsymbol{q})}{\partial \boldsymbol{\theta}} \ . \tag{4}$$

In the limit of small $\boldsymbol{q}$, considering the first order Taylor approximation of the above equation and subtracting equation 3, we obtain:

$$\frac{d\boldsymbol{q}}{dt} = \frac{\partial \left( -\alpha \frac{\partial L(\boldsymbol{\theta})}{\partial \boldsymbol{\theta}} \right)}{\partial \boldsymbol{\theta}} \boldsymbol{q} \ . \tag{5}$$

Then, integrating equation 5, we obtain the evolution of the perturbation under GD:

$$\boldsymbol{q}(t) = \exp \left( -\alpha \frac{\partial^2 L(\boldsymbol{\theta})}{\partial \boldsymbol{\theta}^2} t \right) \boldsymbol{q}_0 \ . \tag{6}$$

This remains true as long as $\boldsymbol{q}(t)$ remains small, where the definition of small depends on the properties of $L$. We consider the decomposition of $\boldsymbol{q}_0$ as a sum of its projections on the eigenspace of the Hessian of the loss (with the Hessian being represented at the exponent of the formula in equation 6). In this space, the projection of $\boldsymbol{q}_0$ along the direction corresponding to the largest eigenvalue is the one growing the fastest. Starting with a random $\boldsymbol{q}_0$, the direction of $\boldsymbol{q}$ that becomes dominant after sufficient time is aligned with the eigenvector of the largest eigenvalue of the matrix at the exponent, and the growth rate of $\boldsymbol{q}$ is equal to the corresponding eigenvalue.

Measuring this growth rate provides a simple and linear (in the number of parameters) method to measure the leading eigenvalue. This procedure represents the calculation of the largest Lyapunov exponent, *i.e.*, the largest eigenvalue ($\lambda_0$) of the matrix $-\alpha \, \partial^2 L / \partial \boldsymbol{\theta}^2$. Due to the minus sign, this corresponds to the smallest eigenvalue ($h_N$) of the Hessian of the loss ($\boldsymbol{H} = \partial^2 L / \partial \boldsymbol{\theta}^2$). More precisely, the smallest eigenvalue of the Hessian and the largest Lyapunov exponent are related as $h_N = -\lambda_0 / \alpha$. For non-convex losses, $h_N$ is the most negative eigenvalue and the matching eigenvector corresponds to the most unstable direction of the optimization of the loss.

Once $\boldsymbol{q}(t)$ is aligned with the largest eigenvector, equation 6 becomes

$$\boldsymbol{q}(t + \Delta t) = \exp(\lambda_0 \Delta t) \boldsymbol{q}(t) \ . \tag{7}$$

The algorithm to calculate $\lambda_0$ requires normalizing the length of $\boldsymbol{q}$ at each step to keep the increment "small". This reference distance is equivalent to the choice of the step size for the calculation of finite difference based second derivative. In dynamical systems terminology this is called calculating the finite-size Lyapunov exponent. Now, the largest Lyapunov exponent is obtained by iterating the following two steps:

$$\lambda_0 \leftarrow \log \left( \frac{\|\boldsymbol{q}(t + \Delta t)\|}{\|\boldsymbol{q}(t)\|} \right) / \Delta t \ , \tag{8}$$

$$\boldsymbol{q}(t + \Delta t) \leftarrow \boldsymbol{q}(t + \Delta t) \frac{\|\boldsymbol{q}(t)\|}{\|\boldsymbol{q}(t + \Delta t)\|} \ ,$$

where $\| \cdot \|$ denotes the L2 norm and $\Delta t$ denotes the time step. One could see that the computation of the largest Lyapunov exponent is analogous to the power method to compute the largest eigenvalue of a given matrix. This idea can be easily extended to compute the top-$k$ Lyapunov exponents following the idea of Benettin et al. (1980). Please refer to the appendix C.

SGD can be described with the same approach, with the loss function replaced by $L(\boldsymbol{\theta}, \boldsymbol{\omega})$ where $\boldsymbol{\omega}$ are random variables that describe which images are picked in each minibatch, the data augmentation used, and in principle any other random process engineered in the network. We note that chaos theory is fully applicable with equivalent results in such general stochastic setting (Arnold, 1988). In the subsequent analysis we will leverage this and work with SGD. Finally, we demonstrate how to extend the method explained in the current section to compute the Lyapunov exponent for SGD with momentum in Appendix B.

## 3    CALCULATING THE LARGEST EIGENVALUE OF THE LOCAL HESSIAN

In section 2.2 we discussed how one can estimate the local largest Lyapunov exponent of the SGD map, which, under some common conditions, corresponds to the most negative eigenvalue of the

---

**Algorithm 1** Computation of the largest eigenvalue of the Hessian

---

**Input** : $L$: loss function, $\mathcal{D}$: training set, $\boldsymbol{\theta} \in \mathbb{R}^N$: point to compute the eigenvalue, $b$: batch size, $\Delta q$: size for Lyapunov exponent

**Output:** $h_0$: the largest eigenvalue of the Hessian of $L$ at $\boldsymbol{\theta}$

$\boldsymbol{q}_0 \in \mathbb{R}^N, \quad \boldsymbol{q}_0 \leftarrow \boldsymbol{q}_0 \frac{\Delta q}{\|\boldsymbol{q}_0\|}$          $\triangleright$ Small perturbation of size $\Delta q$

$\boldsymbol{\theta}_0 \leftarrow \boldsymbol{\theta}, \quad \boldsymbol{\theta}_1 \leftarrow \boldsymbol{\theta} + \boldsymbol{q}_0, \quad \beta \leftarrow 1$          $\triangleright$ Initialization

**while** *not converged* **do**

     $\mathcal{D}_b = \{(\boldsymbol{x}_i, \boldsymbol{y}_i)\}_{i=1}^b \sim \mathcal{D}$          $\triangleright$ Sample a mini-batch

     $\boldsymbol{\theta}_0 \leftarrow \boldsymbol{\theta}_0 + \beta \frac{\partial L(\boldsymbol{\theta}_0; \mathcal{D}_b)}{\partial \boldsymbol{\theta}}$          $\triangleright$ Gradient ascent on $\boldsymbol{\theta}_0$

     $\boldsymbol{\theta}_1 \leftarrow \boldsymbol{\theta}_1 + \beta \frac{\partial L(\boldsymbol{\theta}_1; \mathcal{D}_b)}{\partial \boldsymbol{\theta}}$          $\triangleright$ Gradient ascent on $\boldsymbol{\theta}_1$

     $\lambda \leftarrow \log\left(\frac{\|(\boldsymbol{\theta}_1 - \boldsymbol{\theta}_0)\|}{\Delta q}\right)$          $\triangleright$ Lyapunov exponent iteration

     $\beta \leftarrow \frac{\beta}{\lambda}$          $\triangleright$ Re-scale the learning rate

     $\boldsymbol{\theta}_1 \leftarrow \boldsymbol{\theta} + (\boldsymbol{\theta}_1 - \boldsymbol{\theta}_0)\frac{\Delta q}{\|(\boldsymbol{\theta}_1 - \boldsymbol{\theta}_0)\|}, \quad \boldsymbol{\theta}_0 \leftarrow \boldsymbol{\theta}$          $\triangleright$ Re-centering

**end**

$h_0 \leftarrow \frac{1}{\beta}$          $\triangleright$ Eigenvalue from Lyapunov exponent

---

Hessian of the loss at the same point. While most negative eigenvalues can be used to analyze the existence of saddle points and training instability, in this paper we are interested in computing the largest eigenvalue of the Hessian. Note that the largest eigenvalue corresponds to a theoretical upper bound on the usable learning rate, under quasi-Newton approximation (LeCun et al., 1993). Therefore, by efficiently computing it, we intend to understand the relationship between the SGD optimization trajectory and the learning rate.

To compute the largest (or more generally, the top-$k$) eigenvalues, we need to redefine our map function. The idea is to eliminate the negative sign and use the gradient ascent equation. For this map, the largest Lyapunov exponent corresponds to the largest eigenvalue of the Hessian and the matching eigenvector corresponds to the direction of most instability. We would like to clarify that gradient ascent is used as a map function to estimate the largest eigenvalue of the local Hessian at a particular point in the parameter space. This approach can be employed at any given point, especially at the points in the optimization trajectory, where any algorithm can be chosen for the optimization.

With gradient ascent map, the PDE corresponding to equation 5 can be written as (note the missing minus sign):

$$\frac{\partial \boldsymbol{q}}{\partial t} = \beta \frac{\partial^2 L(\boldsymbol{\theta})}{\partial \boldsymbol{\theta}^2} \boldsymbol{q}, \tag{9}$$

where $\beta > 0$ is the learning rate for gradient ascent (we use a different notation to distinguish it from the SGD learning rate $\alpha$). Similarly, we can integrate equation 9, obtain an exponential form, where the dominating exponential corresponds to the largest eigenvalue. However, this time it corresponds to the largest eigenvalue of the Hessian of the loss denoted by $h_0$.

Since we intend to estimate the Lyapunov exponent at every point in the optimization trajectory, we now discuss how to accelerate the convergence of the Lyapunov exponent computation. To this end, we set up our chaotic map as a control problem, where we optimize the learning rate $\beta$ used for our gradient ascent step such that the convergence of the eigenvector is the fastest but still stable. This is obtained by setting $\beta$ such that the corresponding Lyapunov exponent is controlled to stay close to one. It does not need to be necessarily one, but it needs to be a value of the order of unity. In practice, the learning rate for the next step is re-scaled by the Lyapunov exponent computed in the current step and this numerical procedure ensures that the Lyapunov exponent quickly converges to one.

Our final algorithm to compute the largest eigenvalue of the Hessian at a given point is summarized in Algorithm 1. In practice, this algorithm converges within a couple of iterations and the convergence criterion checks the fluctuation in $\lambda$ around 1. As will be discussed in section 5, in comparison

to (LeCun et al., 1993), our algorithm automatically tunes the learning rate $\beta$ to compute the largest eigenvalue quickly and effectively eliminates one hyper-parameter used in (LeCun et al., 1993). This enables us to run Algorithm 1 at every step to understand the optimization trajectory of SGD and similarly to (LeCun et al., 1993), the largest eigenvalue can be used to automatically set the learning rate of SGD in the quasi-Newton framework.

## 3.1 QUASI-NEWTON METHOD

Quasi-Newton method is an effective approach that utilizes approximate second-order information to improve gradient descent methods. The basic idea is to keep track of an estimate of the Hessian matrix and modify the gradient descent step (Nocedal & Wright, 2006). Formally, at iteration $t$, the quasi-Newton method can be written as:

$$\boldsymbol{\theta}_{t+1} = \boldsymbol{\theta}_t - \boldsymbol{B}_t^{-1} \frac{dL(\boldsymbol{\theta}_t)}{d\boldsymbol{\theta}} , \qquad (10)$$

where $\boldsymbol{B}_t$ denotes the estimate of the Hessian matrix at iteration $t$.

In this paper, we estimate the largest eigenvalue, so the matrix $\boldsymbol{B}_t$ takes the form of $h_0^t \boldsymbol{I}$ where $h_0^t$ is the largest eigenvalue of the Hessian at $\boldsymbol{\theta}_t$ and $\boldsymbol{I}$ is the identity matrix. This is the simplest form of quasi-Newton method which effectively uses $1/h_0^t$ as the learning rate at iteration $t$. This replaces hand-engineered learning rate schedules and could be beneficial when applying SGD to new problems. If top-$k$ largest eigenvalues are estimated as discussed in the appendix, a more sophisticated quasi-Newton approach could be employed.

## 4 EXPERIMENTS

All experiments are based on the CIFAR-10 dataset that has 10 classes with 5000 32×32 pixel training images per class, and 1000 32×32 pixel testing images per class. For all experiments, we use a batch size of $512$ and a weight decay of $5 \times 10^{-4}$ and standard data augmentation techniques. We use a difference of $5 \times 10^{-2}$ ($\Delta q$ in Algorithm 1) in parameter space for calculating the Lyapunov exponent (results do not depend on changing this in a wide range of values). To show that the behavior of the method does not depend on particular loss functions, we run the experiments using the softmax crossentropy and mean square error loss functions. The following models are trained with the first two CIFAR-10 classes (planes and cars): 1) a linear model with mean square error – that is, a least square regression; 2) a Multi-Layer Perceptron (MLP) with three hidden layers; 3) a LeNet1 (LeCun et al., 1998) with relu activations; and 4) a small ResNet with two residual blocks. We also test the larger ResNet18 (He et al., 2016) using all ten CIFAR-10 classes. In all experiments, we use SGD without momentum. One iteration is typically sufficient in the control algorithm to compute $h_0$ at each optimization step, however, noise in $\lambda$ can be alleviated by running more than one iteration for each step.

Figure 1 shows the losses for the MLP training with fixed and cyclic (Smith & Topin, 2019) learning rates. Notice how the inverse of the curvature (orange curve), measured with the controlled Lyapunov exponent, naturally adapts to both the fixed and cyclic learning rates. Since quasi-Newton methods would require the learning rate to be equal or similar to the inverse of the second derivative, we speculate that this discovered behavior is useful to explain the successful training of deep neural networks using first-order methods, such as SGD.

Furthermore, we investigate a hyper-parameter free training based on the measured curvature in Figure 2. We first run our algorithm to measure the curvature on the initialization point, without training. This is to ensure convergence of the eigenvector and avoid a "cold" start. Then, when the starting curvature is known, we set the learning rate to this value and start the training. We keep a simple exponential running average of the curvature to remove noise (this is equivalent to choose a $\Delta t$ in the calculation of the Lyapunov exponent), and set the learning rate (red curve in Fig. 2) to this value dynamically. Empirically, we find that this "optimal" learning rate gradually decreases, guaranteeing a decrease in the loss. We show extensive experiments with analogous results on other architectures in the Appendix D.

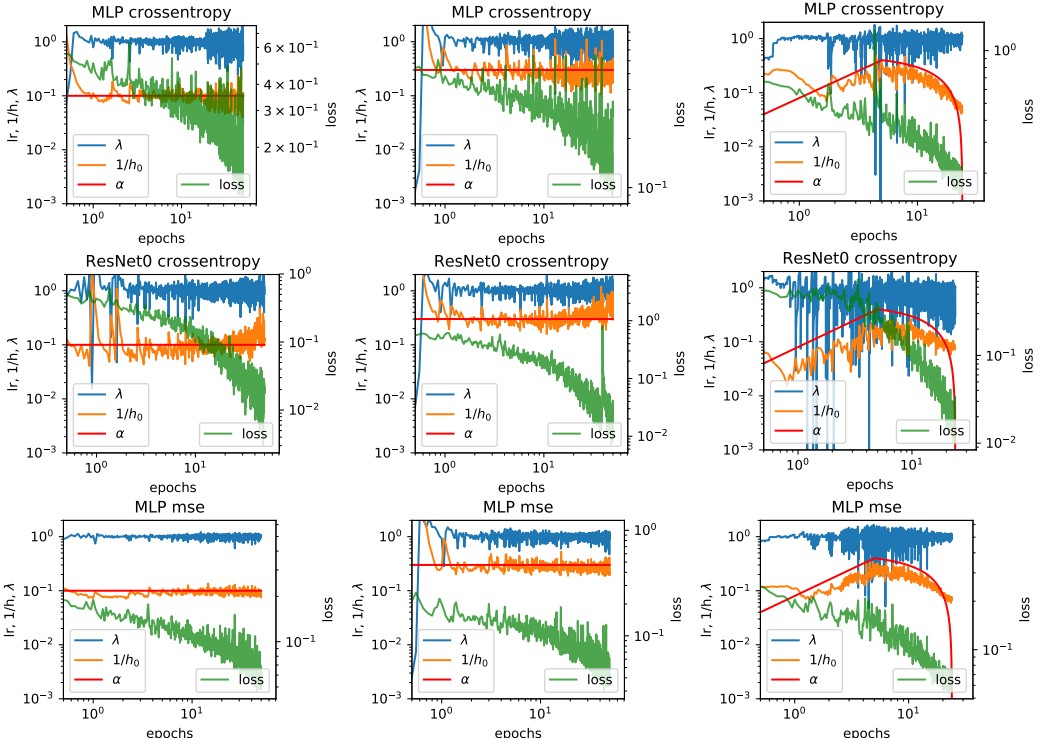

Figure 1: Lyapunov exponent $\lambda$ (blue), inverse curvature from leading eigenvalue $1/h_0$ (orange), learning rate $\alpha$ (red), and loss (green) curves with cross entropy loss for MLP (**top**), tinyResNet (**middle**) and MLP with mean squared error (**bottom**) on a two-class CIFAR10 problem, with constant (0.1, 0.3 – **first two columns**, respectively) and cyclic (**third column**) learning rate schedules. The corresponding experiments on the other setups are in the appendix. Note that $1/h_0$ closely follows $\alpha$ in all cases.

## 5 RELATED WORK

It is interesting how on one hand the design of neural network architectures became progressively more complicated over the years (LeCun et al., 1990; Krizhevsky et al., 2012; Simonyan & Zisserman, 2014; He et al., 2016). But on the other hand, the most popular optimization strategies are still relatively simple and have not substantially changed (Bottou et al., 2018). In fact, the most effective approaches are still based on SGD with minor variations (Ruder, 2016), usually developed to increase their robustness or efficiency. Still, simple SGD often produces state-of-the-art results that are competitive and often even better (Hardt et al., 2016) than more advanced optimization techniques based on second-order methods or pseudo/quasi Newton approaches (Bordes et al., 2009; Sohl-Dickstein et al., 2014) or on adaptive methods (Wilson et al., 2017). Another interesting point about first order methods is their robustness to learning rate choice and schedules (Sun, 2019), evidenced by several methods that study efficient automatic learning rates for SGD (Schaul & LeCun, 2013; Schaul et al., 2013; Li et al., 2017). Hence, there should be some explanation for why simple first-order optimization methods (Ruder, 2016) work so well on large-scale problems with systems containing a large number of parameters, such as deep neural networks.

A similar question has been asked by LeCun et al. (1993), who proposed a method to calculate the largest eigenvalue of the Hessian of the loss using the power method. To make it work along the SGD training, LeCun et al. (1993) designs a running average of the estimate of the eigenvector. Their idea is similar to ours, but our approach has the advantage of being free of the scheduling of the parameter that characterizes the running average. This advantage stems from the Lyapunov exponents analysis presented in our paper, which is one of the first steps in the exploration of the intersection between chaos theory and deep learning. Furthermore, since the largest eigenvalue of the Hessian can be used as an estimate for the smoothness coefficient of the loss function, our approach could improve the

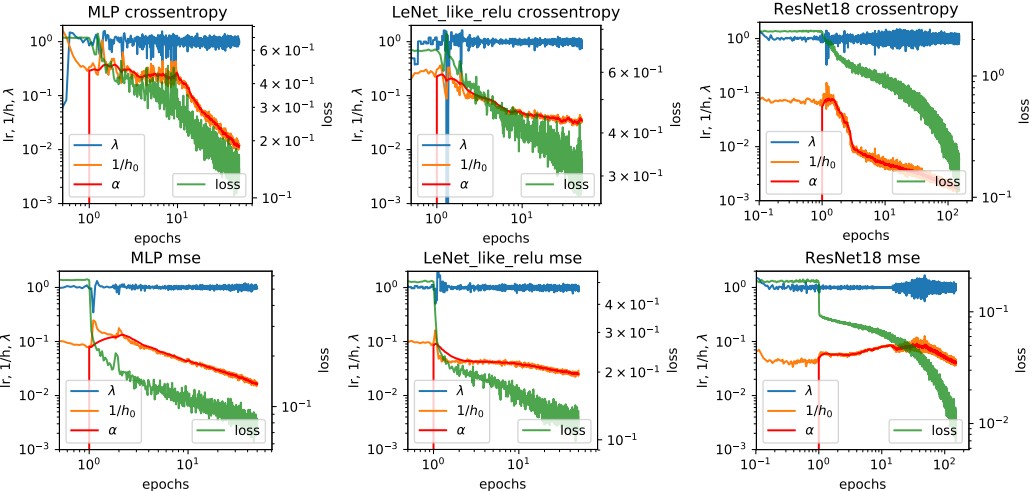

Figure 2: Lyapunov exponent $\lambda$ (blue), inverse curvature from leading eigenvalue $1/h_0$ (orange), learning rate $\alpha$ (red), and loss (green) curves for MLP (**column 1**), LeNet1 (**column 2**), and ResNet18 (**column 3**) with softmax cross entropy loss (**top**) and mean squared loss (**bottom**) on CIFAR10 trained using our quasi-Newton method, *i.e.*, using the estimated $1/h_0$ as the learning rate. While ResNet18 is trained on the full CIFAR10 other networks are trained on a two class CIFAR10 problem. The training begins once the estimation of $h_0$ at the initialization is stable. In all our experiments, the training is stable and loss decreases monotonically.

smoothness coefficient estimation (Santurkar et al., 2018) and help methods that rely on it (Zhang et al., 2019; Lee et al., 2020).

We believe that there are many other topics in this intersection that are worth exploring, such as the use of deep neural networks to predict the behavior of chaotic dynamical systems (Pathak et al., 2018) or the exploration of neural networks as a dynamical system (Liu & Theodorou, 2019; Schoenholz et al., 2016). We align with works that view SGD as an approximation of stochastic differential equations (Chaudhari & Soatto, 2018) or that improve the understanding of empirical and theoretical properties of SGD (Ma et al., 2018; Keskar et al., 2016; Bassily et al., 2020; Kleinberg et al., 2018; Chaudhari & Soatto, 2018), particularly regarding the influence of batch size and learning rate to generalization (He et al., 2019; Jastrzebski et al., 2017; Smith et al., 2017).

## 6 DISCUSSION AND CONCLUSION

In this work, we use a chaos theory approach to design a new method for the efficient estimation of the largest eigenvalue $h_0$ of the Hessian of the loss function. Our proposed method is efficient because it is linear in the number of parameters and can be run in parallel to the optimization. This efficiency allows us to study the dynamical evolution of $h_0$ during training and discover that $1/h_0$ converges to the chosen learning rate $\alpha$. Moreover, we noticed that we could assign $\alpha$ to a large range of values and still have an effective training. Hence, setting the learning rate $\alpha$ with a quasi-Newton optimization is largely superfluous for deep neural networks because of the convergence of $1/h_0$ to $\alpha$. This means that SGD traverses the loss function along a path that has the correct curvature according to second-order optimization. Finally, we have some indications that the convergence of $1/h_0$ towards $\alpha$ is necessary for a successful training. Therefore, our approach could be used to narrow down the initial range of usable learning rates or to design learning rates schedules on new problems.

Although we did not discuss generalization in this paper, we observe that for a fixed batch size, $1/h_0$ follows the learning rate $\alpha$. This means that if larger learning rate is used towards the convergence, a wider optimum will be attained, and wider minima are usually attributed to better generalization (Keskar et al., 2016). This corroborates with previous results that show that the ratio between batch size and learning rate has a negative correlation to generalization (He et al., 2019; Jastrzebski et al., 2017; Smith et al., 2017).

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

## A  APPENDIX

## B  SGD+MOMENTUM

With SGD+momentum the analysis is conceptually similar, but mathematically more complicated.

SGD+momentum:

$$\boldsymbol{\theta}_{t+1} = \alpha \boldsymbol{p}_{t+1} + \boldsymbol{\theta}_t,$$

$$\boldsymbol{p}_{t+1} = m \boldsymbol{p}_t - \frac{\partial L}{\partial \boldsymbol{\theta}}.$$

We can rewrite:

$$\boldsymbol{\theta}_{t+1} - \boldsymbol{\theta}_t = \alpha \boldsymbol{p}_{t+1},$$

$$\boldsymbol{p}_{t+1} - \boldsymbol{p}_t = -(1 - m)\boldsymbol{p}_t - \frac{\partial L}{\partial \boldsymbol{\theta}}.$$

In the limit of small steps:

$$\frac{\partial \boldsymbol{\theta}}{\partial t} = \alpha \boldsymbol{p},$$

$$\frac{\partial \boldsymbol{p}}{\partial t} = -(1 - m)\boldsymbol{p} - \frac{\partial L}{\partial \boldsymbol{\theta}}.$$

Incidentally, in this formulation it becomes clear that $(1 - m)$ is equivalent to a drag term of a particle of mass $\alpha$ under the motion of a potential $L$.

If we define a vector of length $2N$ describing the phase space, where $N$ is the dimension of the parameter space $\boldsymbol{\theta}$ (or $\boldsymbol{p}$):

$$\tilde{\boldsymbol{\theta}} = (\boldsymbol{\theta}, \boldsymbol{p}).$$

The phase space describes both the status of the network and the optimizer's accumulated gradients.

We can rewrite the system of equations above in a compact form:

$$\frac{\partial \tilde{\boldsymbol{\theta}}}{\partial t} = \begin{bmatrix} 0 & \alpha I \\ -\frac{\partial L}{\partial \boldsymbol{\theta}} & -(1 - m)I \end{bmatrix} \tilde{\boldsymbol{\theta}},$$

where $I$ is the identity matrix of size $N$.

Just like in the case of SGD, we obtain the equation for the evolution of a perturbation in the phase space (we call it $\tilde{\boldsymbol{q}}(t)$), and integrate it over $t$ which gives:

$$\tilde{\boldsymbol{q}}(t) = e^{\frac{\partial}{\partial \bar{\boldsymbol{\theta}}} \begin{bmatrix} 0 & \alpha I \\ -\frac{\partial L}{\partial \boldsymbol{\theta}} & -(1-m)I \end{bmatrix}^t} \tilde{\boldsymbol{q}}_0.$$

We need to find the eigenvalues of the matrix at the exponent of the formula, hence we need to solve an equation of the form $|\boldsymbol{A} - \lambda \boldsymbol{I}| = 0$, where $|.|$ represents the determinant:

$$\left| \begin{bmatrix} \boldsymbol{0} & \alpha \boldsymbol{I} \\ -\frac{\partial^2 L}{\partial \boldsymbol{\theta}^2} & -(1-m)\boldsymbol{I} \end{bmatrix} - \lambda \boldsymbol{I} \right| = 0.$$

Rewriting:

$$\left| \begin{bmatrix} -\lambda \boldsymbol{I} & \alpha \boldsymbol{I} \\ -\frac{\partial^2 L}{\partial \boldsymbol{\theta}^2} & (-(1-m) - \lambda)\boldsymbol{I} \end{bmatrix} \right| = 0.$$

The Shur's determinant identity ( $\begin{vmatrix} \boldsymbol{A} & \boldsymbol{B} \\ \boldsymbol{C} & \boldsymbol{D} \end{vmatrix} = |\boldsymbol{D}||\boldsymbol{A} - \boldsymbol{B}\boldsymbol{D}^{-1}\boldsymbol{C}|$ ) gives:

$$\left| -\lambda(-(1-m) - \lambda)\boldsymbol{I} - \alpha \left( -\frac{\partial^2 L}{\partial \boldsymbol{\theta}^2} \right) \right| = 0,$$

which is a formula of the form $|\boldsymbol{H} - h\boldsymbol{I}| = 0$. This means that the eigenvalues of the hessian of the loss ($h$) are related to the Lyapunov exponents by the formula:

$$h = -\frac{\lambda^2 + \lambda(1-m)}{\alpha}.$$

Similarly to the SGD case, the largest $\lambda$ gives the smallest $h$. The final formula in this case becomes:

$$h_N = -\frac{\lambda_0^2 + \lambda_0(1-m)}{\alpha}.$$

## C  LYAPUNOV EXPONENT SPECTRUM

Our curvature estimation idea can be easily extended to estimate the top-$k$ (negative or positive) eigenvalues of the Hessian. We calculate the first Lyapunov exponents using the orthogonalization procedure described by Benettin et al. (1980). To calculate the second Lyapunov exponent, it is enough to keep track of a second "small" increment vector $\boldsymbol{q}^{(1)}(t)$ evolved in exactly the same way as $\boldsymbol{q}(t)$, with an additional orthogonalization step (Benettin et al., 1980):

$$\boldsymbol{q}^{(1)}(t + \Delta t) \leftarrow \boldsymbol{q}^{(1)}(t + \Delta t) - \frac{\boldsymbol{q}^{(1)}(t + \Delta t) \cdot \boldsymbol{q}(t + \Delta t)}{\|\boldsymbol{q}(t + \Delta t)\|}$$

to be done before the corresponding normalization step. The procedure is easy to generalize to further eigenvalues.

The results typically show a shallow dependence of the value with eigenvalue number (Figure 3). If this holds true in general, calculating additional eigenvalues will be of limited usefulness for improving optimization.

## D  ADDITIONAL EXPERIMENTS

In this section we present additional experiments of the same type shown in section 4 done with different architectures and loss functions. Figures 4,5 and 6 show experiments with constant learning rate. Figure 7 shows experiments with cyclic learning rate. And Figures 8 and 9 show the full set of experiments on two classes CIFAR10 with our quasi-newton method for SGD. It is worth mentioning how it trains also with the linear model regression (Figure 9, upper-left). Figure 10 shows experiments with the larger ResNet18 architecture and the 10 classes of CIFAR10. The same behavior as in the main paper is consistently observed across different architectures/losses. It is possible to mitigate the noise in $\lambda$, and consequently in $1/h_0$, by increasing the number of iterations of Algorithm 1. As explained in the main text, $1/h_0$ cannot follow $\alpha$ when it is too small (e.g. Fig 7), that is, the curvature cannot go to infinity.

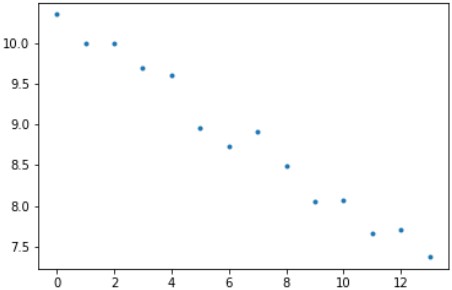

Figure 3: The first 14 eigenvalues of the Hessian of the loss averaged over the first epoch.

Figure 4: MLP, LeNet1, tinyResNet with a mean square error and soft-max cross entropy loss trained on a two classes cifar10 with a constant learning rate (0.05). The linear model regression does not train in this setup.

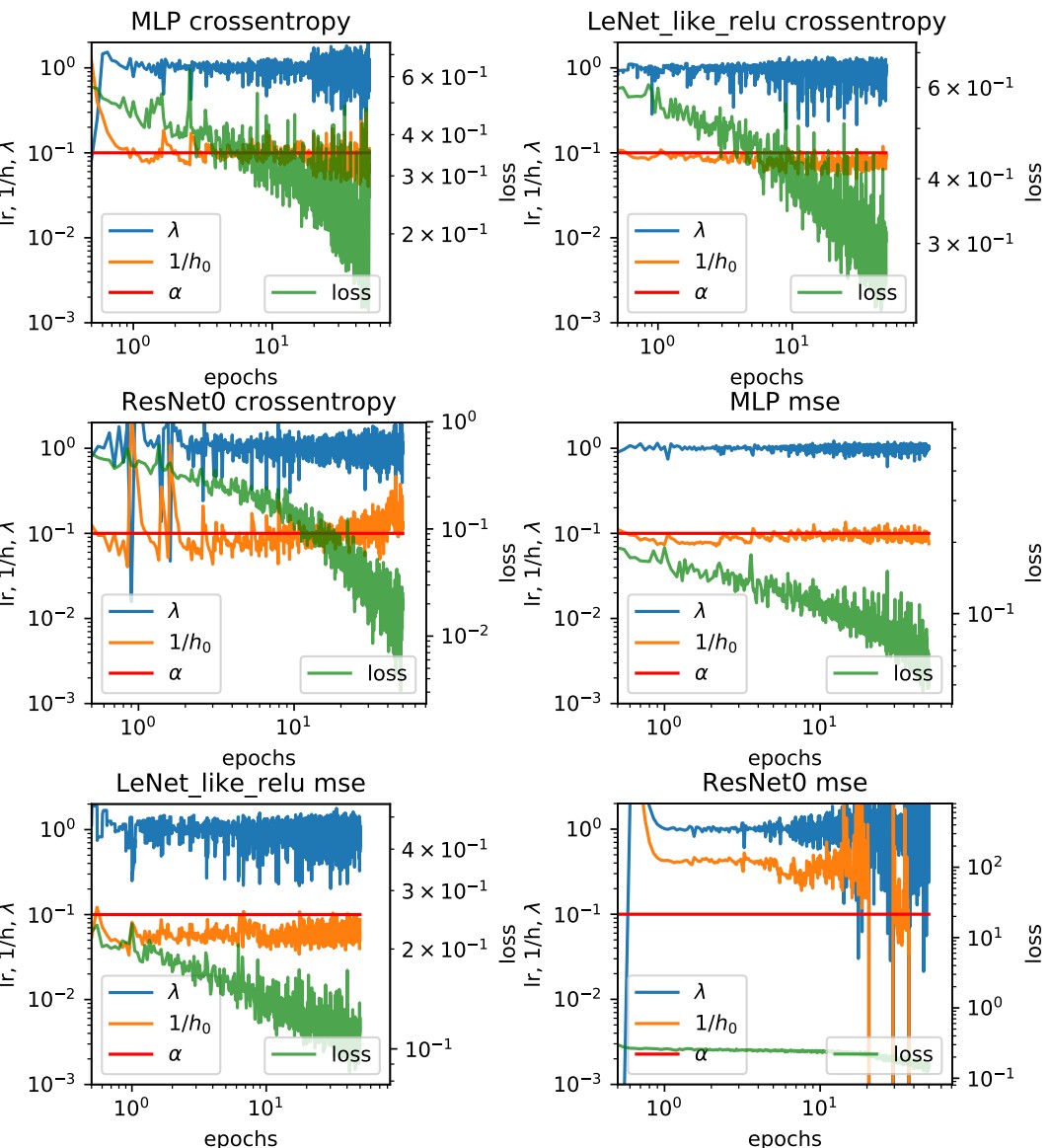

Figure 5: MLP, LeNet1, tinyResNet with a mean square error and soft-max cross entropy loss trained on a two classes cifar10 with a constant learning rate (0.1). The linear regression does not train in this setup.

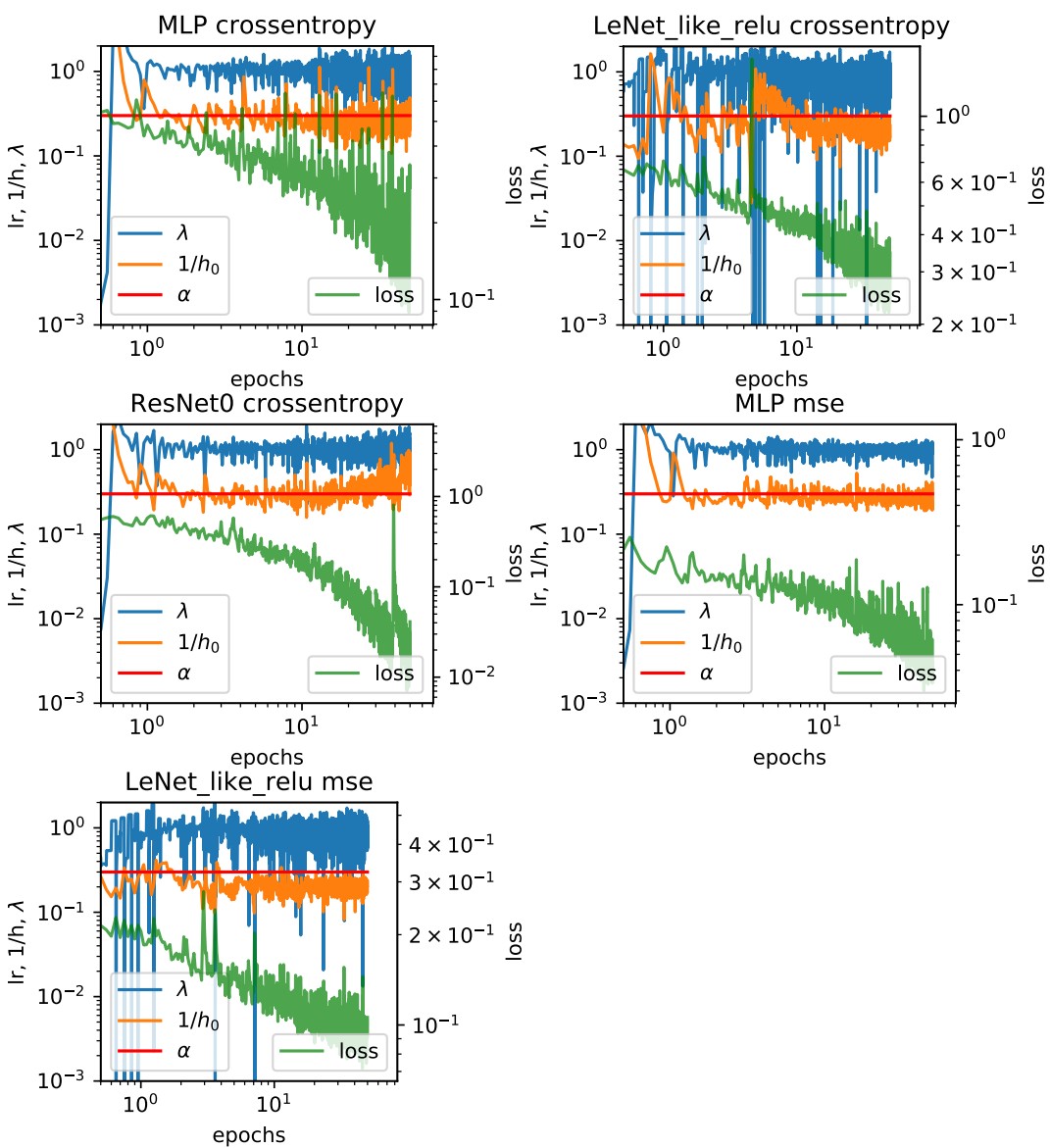

Figure 6: MLP, LeNet1, tinyResNet with a mean square error and soft-max cross entropy loss trained on a two classes cifar10 with a constant learning rate (0.3). The linear regression and the ResNet with MSE do not train with this setup.

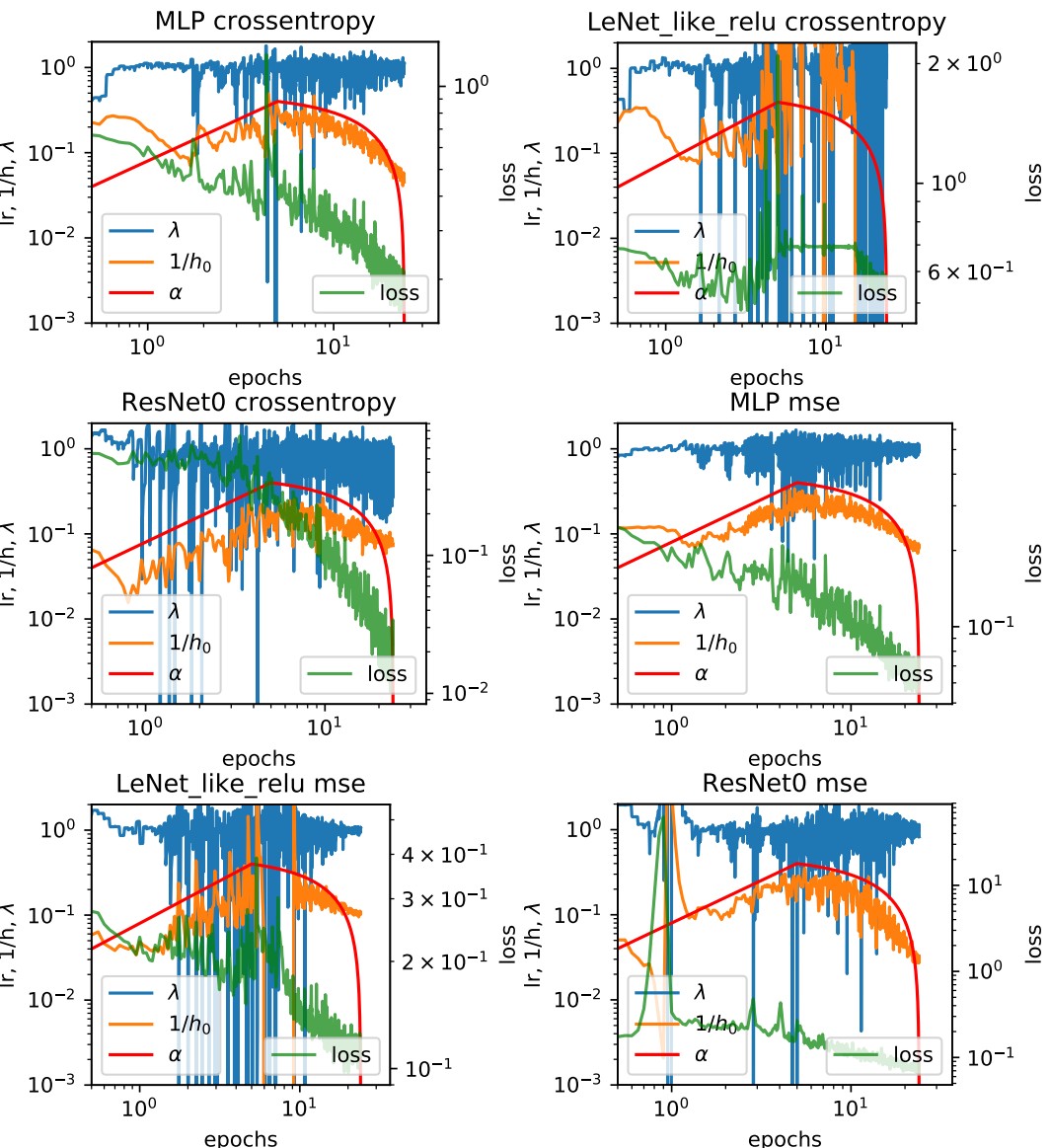

Figure 7: MLP, LeNet1, tinyResNet with a mean square error and soft-max cross entropy loss trained on a two classes cifar10 with a cyclic learning rate. The linear model regression does not train in this setup.

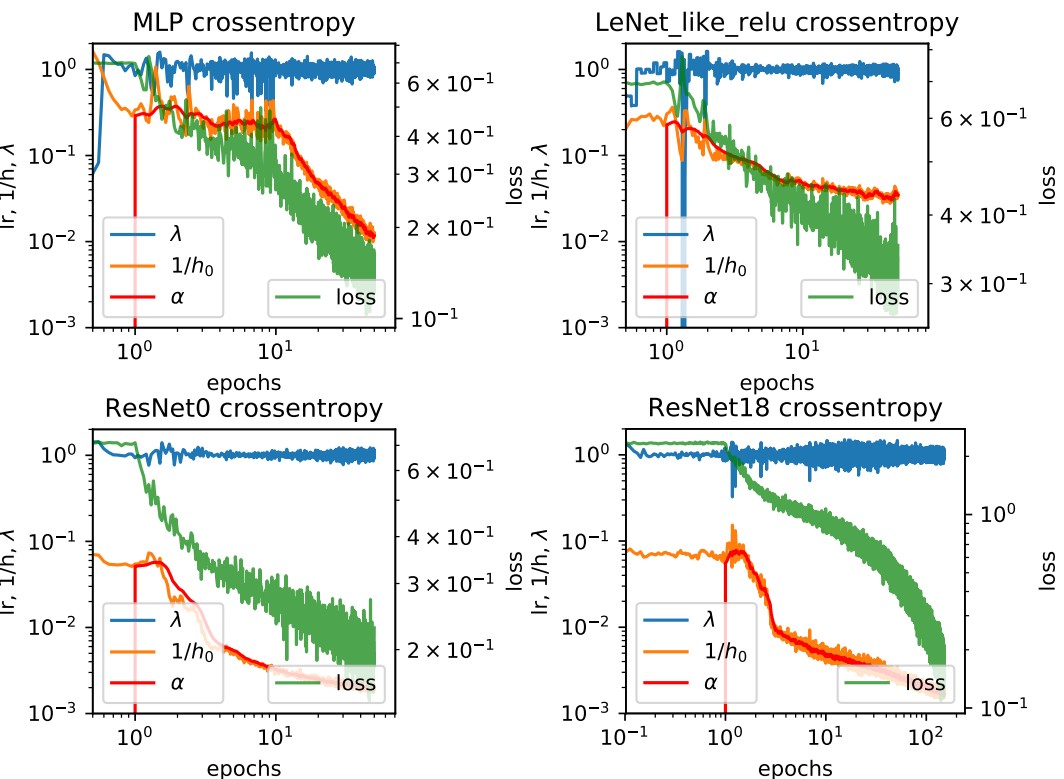

Figure 8: Lyapunov exponent $\lambda$ (blue), inverse curvature from leading eigenvalue $1/h_0$ (orange), learning rate $\alpha$ (red), and loss (green) curves for MLP, LeNet1, tinyResNet trained with a softmax-crossentropy loss on a two-class CIFAR10 problem; and ResNet18 trained on all ten CIFAR10 classes. All the trainings are with the quasi-Newton method based on method to calculate the largest eigenvalue of the Hessian of the loss.

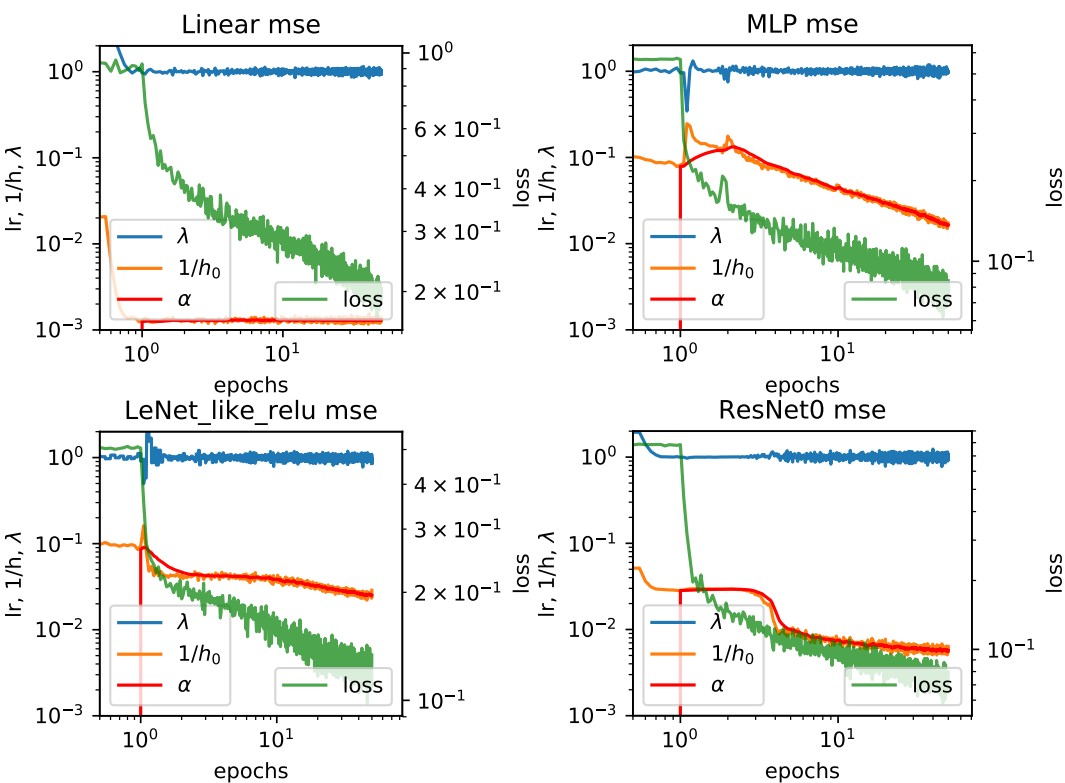

Figure 9: Lyapunov exponent $\lambda$ (blue), inverse curvature from leading eigenvalue $1/h_0$ (orange), learning rate $\alpha$ (red), and loss (green) curves for a linear model, MLP, LeNet1, tinyResNet with a mean square error loss trained on the two-class CIFAR-10 problem with a quasi-Newton method based on method to calculate the largest eigenvalue of the Hessian of the loss. Notice how the linear least square regression can be successfully trained.

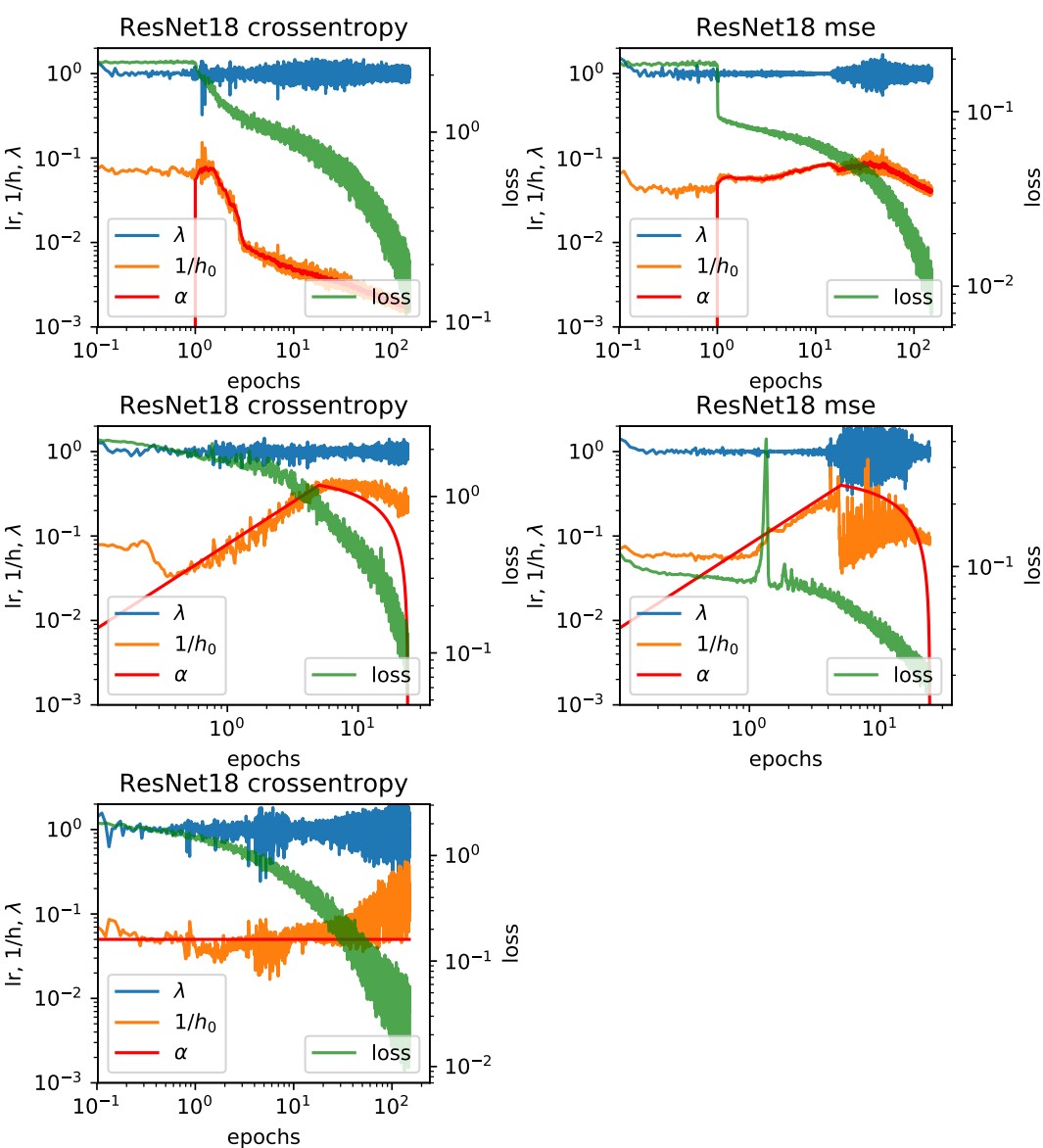

Figure 10: ResNet18 with soft-max cross entropy loss trained on a cifar10 with our quasi-newton method, a cyclic lr and constant lr of $0.05$.

