# OpenReview forum: "A Chaos Theory Approach to Understand Neural Network Optimization"
_ICLR.cc/2021/Conference — Reject_

### Official Review · AnonReviewer3 · 2020-10-27
**Official Blind Review #3**

**Rating:** 4
**Confidence:** 3

**Review:**

Interesting approach using ideas in chaos theory to deep learning but its merit is not clear yet.

Summary:

This paper connects ideas in chaos theory to understand training dynamics of neural networks. In particular the paper uses Lyapunov exponent, divergence rate of infinitesimal close trajectory and connects them to Hessian eigenvalues. The authors show that the largest Lyapunov exponent corresponds to the most negative eigenvalue of the Hessian. Then the paper claims that this provides an efficient method to estimate the largest eigenvalue and connect to using learning rate related to largest eigenvalue.

Using this method, the paper claims SGD finds loss landscape regions where the largest eigenvalue of the Hessian is similar to the inverse of the learning rate. Lastly, the paper proposes a quasi-Newton method with dynamic estimation of optimal learning rate.


Reason for score:

While connection between ideas in chaos theory and neural network optimization is interesting and worth pursuing, I believe current work is underdeveloped in the sense that comparison to well-known methods are not provided thoroughly. Proposed method of estimating largest eigenvalue of Hessian is presented without any comparison to well-known methods such as Lanczos and proposed quasi-Newton method’s utility is unproven as is.


Pros:

The paper proposes an interesting connection between analysis in chaos theory and neural network optimization.

Paper is clearly written and exposition to chaos theory in section 2 is  a nice read.

Empirical observation that maximum eigenvalue of Hessian over training matching learning rate for various experimental settings is quite interesting. While I would have been interested to see similar observations for commonly used step learning rate schedules.


Cons:

The analysis is based on continuous time dynamics of SGD which is a fine toy-model but misses various interesting finite step dynamics in Neural Network training. For example [1] have shown that the largest learning rate based on eigenvalue estimation is not sufficient for explaining neural network training dynamics especially in the well-performant ones.

One major problem I see is missing comparison to well known literature. There are numerous analyses for studying Hessian of Neural Networks with various methods(e.g. [2, 3, 4] and references there-in), and it is hard to find why the proposed method using the Lyapunov exponent is either more interesting or useful.

For one thing there are various methods to estimate large eigenvalues of large matrices. For example, naively I would have used Lanczos to estimate the top few eigenvalues very efficiently using Hessian-Vector product. Why would a proposed method be better than this?

According to discussion in Related work section, the benefit over power-iteration from (LeCun et al., 1993) is that it is free from choosing a running average hyperparameter, in which case the novelty of the proposed method itself does not seem significant especially with lack of analysis that directly compares one another.

Lastly, while the authors suggest that the method is efficient, most experiments are done in the toy-ish setting whereas [3] could study full Hessian spectral density of ImageNet-scale networks.


Proposed quasi-Newton does not have sufficient analysis that the idea works. There are numerous ideas for proposing new optimization and without careful, through comparison to baseline, well-known methods. I believe Figure 2 is testing that the optimization works. However it is not clear with the current set of experiments whether this optimization can be useful compared to simple Adam or SGD with momentum.

While one could eliminate the learning rate schedule with this method, it is not clear whether the proposed quasi-Newton method provides benefit over used schedules. SGD works well without schedule, but typically schedule improves performance beyond constant learning rate. Does automatic determination of learning rate provide the benefit of custom learning rate schedule without any tuning procedure? I think this question needs to be answered for the proposed method to have impact on practitioners.



Questions:

As far as I can see, the procedure for both top eigenvalue and top-k eigenvalue is very similar to how one would estimate them using simple power iteration or Lanczos algorithm(https://en.wikipedia.org/wiki/Lanczos_algorithm) used in e.g. [2,3,4]. Could you explain how they differ? At least stochastic estimation (not using full batch) has been utilized in [3] where they study dynamics of Hessian spectral density during Training and I believe few top value estimation is much simpler to extract.

Do the authors believe chaos in parameter space is a relevant question to answer? In the end, one is interested in neural network function and even if the parameters diverge the function output may converge since many different parameter configurations can lead to the same or similar functions.

In section 6, suggestion for using larger learning rate towards the end of training seems to be against the typical practice for obtaining well performing models. I believe even suggested reference (Smith et al., 2017) suggests decaying the learning rate is a good idea for generalization and mimics the effect by increasing the batch size.

Would be interesting to find out if the analysis in Appendix B can generalize to Adaptive optimization algorithms such as RMSProp or Adam.



Nits and additional feedback:

These are few nits and feedback to improve the paper which were not critical for evaluation:

Ref (Sprott & Sprott 2003) seems to be actually a single author book, I suspect bibtex is misconfigured.

For the experiments section (4), it is not clear what message the experiments are conveying. The experiment setup without knowing what motivates the analysis is hard to follow, so I suggest starting with a general goal for the experiments to help the readers.

For second-order methods in Neural Networks in the related works section, it is worth mentioning K-FAC papers [5,6,7]

I am not sure if the statement “robustness to learning rate choice” is correct in Section 5. In most cases, choosing a proper learning rate for a first order method is quite important and probably the single most important hyperparameter to tune. If the learning rate is too small, convergence will be too slow, if it is too large SGD can diverge. Also there’s evidence that a very small range of learning rate is critical for improving performance [1].

Figure 3 with x, y axis labels missing

Inconsistent use of cifar10, CIFAR10, CIFAR-10 across the paper

Understandable for conference submission with deadlines; content in the Appendix needs more cleaning up. (e.g. wrong quotation marks, ‘newton’ instead of ‘Newton’ etc)


[1] Lewkowycz et al., The large learning rate phase of deep learning: the catapult mechanism, arXiv:2003.02218
[2] Gur-Ari et al., Gradient Descent Happens in a Tiny Subspace, arXiv:1812.04754
[3] Ghorbani et al., An Investigation into Neural Net Optimization via Hessian Eigenvalue Density, ICML 2019
[4] Jastrzebski et al., On the Relation Between the Sharpest Directions of DNN Loss and the SGD Step Length, ICLR 2019
[5] Martens & Grosse, Optimizing Neural Networks with Kronecker-factored Approximate Curvature, ICML 2015
[6] Grosse & Martens, A Kronecker-factored approximate Fisher matrix for convolution layers, ICML 2016
[7] Ba et al.,  Distributed second-order optimization using Kronecker-factored approximations. ICLR 2017.

---

> ### Author Response · Authors · 2020-11-13
> **reply to review**
>
> Replies to "cons":
> - The analysis is presented using continuous time dynamics of SGD for simplicity of exposition. All the concepts used (chaos, Lyapunov exponents) can be derived and applied on discrete time systems.
> - We will add these to the literature discussion.
> - The Lanczos method is an extension to the power method to calculate multiple eigenvalues. As explained in the appendix, what we are doing can also be extended to multiple eigenvalues.
> - We disagree on the significance of the novelty.
> - Our biggest networks are not toy models. The biggest network is a ResNet18 architecture. Our results apply to the simple Linear mean square error fit, MLPs, up to deep ResNet architectures.
> - With figure 2 show how this simple method allows a decrease in the loss for any architecture choice and no need for LR fine-tuning. We do not claim it to be competitive to SGD+momentum/Adam.
>
> Replies to "questions":
> - Our method is a power method where the rate of convergence (Lyapunov exponent) is controlled to be one (“Re-scale the learning rate” step in Algorithm 1). Doing this avoids slow convergence (small lambda) and instability (large lambda).
> - We believe that investigating the chaotic behavior in parameter space is a very relevant question. For example, the structure of the attractor determines the properties of the ensemble of all possible solutions. And in case of chaotic systems, this structure is often non intuitive.
> - We will amend this argument in the revision.
> - The analysis in appendix B can in principle be generalized to other algorithms.

---

### Official Review · AnonReviewer4 · 2020-10-28
**A review**

**Rating:** 5
**Confidence:** 3

**Review:**

### 1. Brief summary
The authors use an insight from chaos theory to derive an efficient method of estimating the largest and smallest eigenvalues of the loss Hessian wrt the weights. To do that, they use nearby weight space positions, optimize for a bit (either gradient climbing or descending), check how quickly the points are departing from each other, and use that to estimate the extreme eigenvalues using a connection to Lyapunov coefficients in chaos theory. Then they use on the fly estimated largest eigenvalue to automatically tune the learning rate of SGD.

### 2. Strengths
* The paper makes a connection to chaos theory which typical members of the ML community are not familiar with
* They derive an alternative to the usual top and bottom eigenvalue calculation methods that are employed
* They try their automatic LR tuning in practice

### 3. Weaknesses and points of confusion

1) The only dataset tested was CIFAR-10. I am not saying you need to go directly to ImageNet, but a variety of datasets would be nice to see. You do try a bunch of architectures, so why not datasets as well. You could add MNIST, Fashion MNIST and SVHN relatively quickly and it would greatly strengthen the empirical conclusions.

2) The simplest method for estimating the top eigenvalue -- the power method -- is also linear in the number of parameters. What advantage does your method have over that?

3) The power method tends to be unstable (in its naive implementation) when used to get the less than highest eigenvalues. Does your method suffer from similar practical instabilities?

4) The connection between the top negative eigenvalue and the rate of departure of nearby points in the weight space from each other (the same for gradient ascent and the top eigenvalue) does not seem very surprising to me. This might not be a valid point, but it seems that it is a simple consequence of optimizing in a quadratic well with a loss of the form 1/2 x H x^T, where H is the Hessian and the x is the minimum-centered position. The highest negative eigenvalue will be the one pushing you out as exp(|lambda| t). Why do I need chaos theory to see that? I might be wrong and I'm ready to be corrected, but it seems relatively simple to derive without much chaos-theoretic baggage attached to it.

5) Almost every stability analysis of gradient-based algorithms will include the condition on the top eigenvalue being smaller than 2/LR and a very similar analysis to what you did using chaos theory here. I'm not sure what the new insight is here. Again, please correct me if I'm wrong.

6) It seems that what you are describing with your adaptive optimization is very similar to some existing algorithms. [1] presents the lookahead optimizer that shares many features, and many variants of SGD (such as SGD+Momentum or Adam) likely do something very similar albeit implicitly.

7) In Equation 10 you make the B a matrix, but it turns out to be an identity rescaled by the top eigenvalue -- a scalar. I get that this is the same, but it seems a bit misleading -- I got my hopes up for a proper matrix conditioning the LR but it turned out to be a scalar. This is a minor point, no need to address it.

8) You argue that the 2/LR top eigenvalue selection by the optimizer somehow helps explain why DL works so well. But to me the more interesting questions remain: why are such places available, and how are they reacheable from init using gradient-based algorithms.

### 4. Some papers that seem relevant
[1] Lookahead Optimizer: k steps forward, 1 step back. Michael R. Zhang, James Lucas, Geoffrey Hinton, Jimmy Ba https://arxiv.org/abs/1907.08610

[2] Deep Ensembles: A Loss Landscape Perspective by Stanislav Fort, Huiyi Hu, Balaji Lakshminarayanan (https://arxiv.org/abs/1912.02757) studies how trajectories in the weight space diverge from different initializations.

[3] Linear Mode Connectivity and the Lottery Ticket Hypothesis by Jonathan Frankle, Gintare Karolina Dziugaite, Daniel M. Roy, Michael Carbin (https://arxiv.org/abs/1912.05671) looks at how trajectories that start from a preoptimized point diverge with additional training.

[4]  Deep learning versus kernel learning: an empirical study of loss landscape geometry and the time evolution of the Neural Tangent Kernel by Stanislav Fort, Gintare Karolina Dziugaite, Mansheej Paul, Sepideh Kharaghani, Daniel Roy, Surya Ganguli (https://arxiv.org/abs/2010.15110 and NeurIPS 2020) also looks at how trajectories from nearby points diverge. They also look at the sensitivity to initial conditions.

[5] The large learning rate phase of deep learning: the catapult mechanism by Aitor Lewkowycz, Yasaman Bahri, Ethan Dyer, Jascha Sohl-Dickstein, Guy Gur-Ari studies the stability of the training under finite step size and with SGD in quite some detail and it could be relevant. (https://arxiv.org/abs/2003.02218)

[6] The Break-Even Point on Optimization Trajectories of Deep Neural Networks by Stanislaw Jastrzebski, Maciej Szymczak, Stanislav Fort, Devansh Arpit, Jacek Tabor, Kyunghyun Cho, Krzysztof Geras (https://arxiv.org/abs/2002.09572 and ICLR 2020) looks at the crucial effect of the early stages of training and the instability in it.

### 5. Summary
This paper presents a nice new method for estimating the lowest and largest eigenvalues of the DNN loss Hessian wrt weights using the divergence of nearby points in the weight space under optimization. They do this by using a chaos-theoretic language. While those methods seem useful, I do not see why chaos theory was needed to derive them. I appreciate the link and believe that more good stuff could come out of it, but as is I don't think this paper provides much new to the field on its own. However, I am not an expert on this subfield and **I am ready to revise my score** if the authors convince me otherwise.

---

> ### Author Response · Authors · 2020-11-13
> **reply to weaknesses**
>
> Reply to weaknesses:
> 1) CIFAR-10 was used in two configurations: 2 classes and 10 classes classification. Additional datasets could be added quickly. We kept it concise for the sake of the space limits. However, we do not think that they would strengthen the conclusions greatly.
> 2) The calculation of the largest Lyapunov exponent (section 2.2) is a power method. Our contribution (explained in section 3) of controlling the Lyapunov exponent to be unity avoids slow convergence (Lyapunov exponent too small) and instability (Lyapunov exponent too large).
> 3) We did not investigate extensively calculating multiple eigenvalues. The stability benefits should apply to this scenario as well.
> 4) The reviewer is correct on this simple case. However, there are a number of non intuitive behaviours  of chaotic dynamical systems that would be very difficult to “rediscover” from scratch.
> 5) What we found in addition is that the eigenvalues become approximately equal to the inverse of the LR.
> 6) Our adaptive optimization is a quasi-netwon method based on the eigenvalue of the Hessian. SGD+momentum and Adam are not based on the eigenvalues of the Hessian.
> 7) If more than one eigenvalue is used, the LR can be made dependent from the direction.
> 8) We believe the techniques described here have a great potential for understanding of NN optimization these additional interesting questions.

---

### Official Review · AnonReviewer2 · 2020-10-29
**A way to compute the top eigenvalues of Hessian from Lyapunov exponents**

**Rating:** 4
**Confidence:** 5

**Review:**

Objective of the work: The paper uses the chaotic theory to study the dynamics of SGD. It provides algorithm to compute the most positive and the most negative eigenvalues of Hessian based on analyzing the Lyapunov exponents. The paper shows that the largest eigenvalue of the Hessian similar to the inverse of the learning rate.

Strong points: The paper proposes an algorithm that can fast estimate the largest eigenvalues of the Hessian based on the analysis of the Lyapunov exponents. The technical derivation is sound.

Weak points:
1. Chaos theory provides a way of computing eigenvalues but does not give much understanding on the neural network optimization. For example, what does the timescale of the most negative eigenvalue mean for the NN optimization.

2. There are several points the paper wants to present, however they are not logically connected: the most negative eigenvalue, the largest eigenvalue of the Hessian, the relation between the largest eigenvalue of Hessian and the learning rate.
The experiments show that the eigenvalues of the Hessian adapts to the learning rate, which indicates the learning rate sort of affects the Hessian, which indicates that we should not follow the largest Hessian eigenvalue if we want certain feature, i.e., large eigenvalue for a wide valley. However, the paper also propose setting the learning rate according to the Hessian leading eigenvalues. Is there first eigenvalue or first the learning rate.


3. For computing the eigenvalues of Hessian, the paper does not give sufficient discussion or experiments of comparing the proposed algorithms with other existing approaches (Hessian vector product method, the density method [1]) to verify the efficiency and the difference.

I would not recommend the acceptance for now.

[1] Ghorbani et al. An Investigation into Neural Net Optimization via Hessian Eigenvalue Density, ICML 2019

---

> ### Author Response · Authors · 2020-11-13
> **Clarifications**
>
> We thank the reviewer for praising that the technical derivation is sound.
> Replies to the weak points:
> 1) The inverse of the most negative eigenvalue of the Hessian defines the chaotic timescale of the training dynamics. In the training of typical neural networks this time is short. From just a few iterations up to an epoch. The consequences of this behavior to NN optimization is certainly an interesting question. We do not see this as a weak point, we just decided to not focus on this topic in our current submission.
> 2) A large value for the first eigenvalue of the Hessian corresponds to a narrow valley. The reviewer correctly points out that our analysis shows that choosing (imposing) a learning rate affects the “width” of the solution. On the other hand, the Hessian can be used to calculate the locally “optimal” learning rate by Quasi-Newton. If this is done instead, the “width” of the final solution will be different. There is no tension between these two observations.
> 3) The Hessian vector product method returns a derivative from automatic differentiation. Our method returns finite difference second derivatives. In networks with relu activations it can make a significant difference. The density method calculates an estimate of the distribution of all the eigenvalues. Our method returns the largest one.

---

### Decision · Program_Chairs · 2021-01-07
**Final Decision**

**Decision:**

Reject

**Comment:**

The Authors study the learning dynamics of deep neural networks through the lenses of chaos theory.

The key weakness of the paper boils down to a lack of clarity and precision. Chaos theory seems to be mostly used to computing eigenvalues but is not used to derive meaningful insights about the learning dynamics. R2 noted, "Chaos theory provides a way of computing eigenvalues but does not give much understanding on the neural network optimization.". R4 noted, "The authors use an insight from chaos theory to derive an efficient method of estimating the largest and smallest eigenvalues of the loss Hessian wrt the weight". Hence, statements such as "the rigorous theory developed to study chaotic systems can be useful to understand SGD" seem unsubstantiated.

Reduced to its essence, the key contribution is (1) a method to compute the top and the smallest eigenvalue, (2) the observation that the spectral norm of the Hessian along SGD optimization trajectory is related to the inverse of the learning rate, and (3) a method to automatically tune the learning rate.

Let me discuss these three contributions:

* The significance of the first contribution is unclear, as pointed out by R2. Indeed there are other methods (e.g. power method, Lanczos) for computing these quantities that should achieve either a similar speed or similar stability. Given the rich history of developing estimators of these quantities, a much more detailed evaluation is warranted to substantiate this claim.

* The core insight that the top eigenvalue of the Hessian in SGD is related to the inverse of the learning rate in the training of deep neural networks is nontrivial but is not fully novel. Closely related observations were also shown in the literature.
This precise statement however indeed was not stated in the literature. This contribution could be a basis for acceptance, but the paper is not sufficiently focused on it, and the evaluation of this claim is a bit narrow in scope.

* Finally, there is a range array of methods to tune the learning rate. As noted for example by R3, "There are numerous ideas for proposing new optimization and without careful, through comparison to baseline, well-known methods", the evaluation is too limited to treat this as a core contribution.

Based on the above, I have to recommend the rejection of the paper. At the same time, I would like to thank the Authors for submitting the work for consideration to ICLR. I hope the feedback will be useful for improving the work.